# Exploiting somatic piRNAs in *Bemisia tabaci* enables novel gene silencing through RNA feeding

Mosharrof Mondal[1], Judith K Brown[1], Alex Flynt[2]

**RNAi promises to reshape pest control by being nontoxic, biodegradable, and species specific. However, due to the plastic nature of RNAi, there is a significant variability in responses. In this study, we investigate small RNA pathways and processing of ingested RNAi trigger molecules in a hemipteran plant pest, the whitefly *Bemisia tabaci*. Unlike *Drosophila*, where the paradigm for insect RNAi technology was established, whitefly has abundant somatic piwi-associated RNAs (piRNAs). Long regarded as germline restricted, piRNAs are common in the soma of many invertebrates. We sought to exploit this for a novel gene silencing approach. The main principle of piRNA biogenesis is the recruitment of target RNA fragments into the pathway. As such, we designed synthetic RNAs to possess complementarity to the loci we annotated. Following feeding of these exogenous piRNA triggers knockdown as effective as conventional siRNA-only approaches was observed. These results demonstrate a new approach for RNAi technology that could be applicable to dsRNA-recalcitrant pest species and could be fundamental to realizing insecticidal RNAi against pests.**

## Introduction

RNAi technology has been shown to be applicable as a low-toxicity biopesticide to control agricultural insect pests and vectors of plant pathogens through silencing essential, biologically relevant genes (Zotti & Smagghe, 2015). RNAi shows great potential to be highly species specific and thereby spares beneficial organisms and is nontoxic to humans and other animal consumers. The RNAi approach for insect pest/vector control relies on the ingestion of long dsRNAs to trigger gene silencing via siRNA production after Dicer processing (Head et al, 2017; Knorr et al, 2018). Although a number of products are available, some arthropod pests exhibit moderate or only minor sensitivity to dsRNA upon ingestion (Yu et al, 2013; Zhu & Palli, 2020). This suggests that to fully realize this strategy across most or all arthropods, RNAi triggers may require unique engineering relevant to each target species (Shukla et al, 2016; Parsons et al, 2018).

Here, RNAi pathways were investigated for the whitefly *Bemisia tabaci* (Genn.) (Aleyrodidae, Hemiptera) to characterize the fundamental features that might be exploited to improve RNAi approach(es). As a group, this whitefly is considered a cryptic or sibling species. Although most *B. tabaci* are relatively benign, at least two variants/cryptic species transmit plant viruses and are among the most invasive species causing damage to crops grown in subtropical, tropical, and mild temperate parts of the world (Brown et al, 1995; Brown, 2010; Chen et al, 2016; de Moya et al, 2019; Grover et al, 2019). Chemical pesticides can be toxic to the environment and consumers of these products and regularly have become ineffective when resistance develops (Chen et al, 2016). *B. tabaci* is closely related to greenhouse and spiraling whiteflies, and several other related phloem-feeding pests/pathogen vectors, including aphids, mealybugs, and psyllids. Previous RNAi studies with *B. tabaci* have demonstrated gene silencing in response to long dsRNA feeding; however, processing modes of these molecules and those in other non-holometabolous insects have not been characterized at the level of small RNA effector populations (Jaubert-Possamai et al, 2007; Zha et al, 2011; El-Shesheny et al, 2013; Thakur et al, 2014; Li et al, 2016; Wang et al, 2016; Vyas et al, 2017; Grover et al, 2019; Kanakala et al, 2019). In this study, the behavior of these molecules is investigated in the context of an extant RNAi mechanism in *B. tabaci*.

Multiple biogenesis modes are reported for animal small RNAs, generally though three main classes are recognized: miRNAs, siRNAs, and Piwi-associated RNAs (piRNAs) (Carthew & Sontheimer, 2009). miRNAs are deeply conserved, short hairpin––derived RNAs that are present in the cells of nearly all metazoans (Bartel, 2018). In contrast, the biology of the other two small RNA classes is highly variable, likely due to their role in defense against invasive nucleic acids like transposable elements (TEs) and viruses (Okamura, 2012). Indeed, their overall roles in animals have been observed to diverge even among family members (Ozata et al, 2019). Each small RNA variety is sorted to distinct argonaute (Ago)/Piwi proteins with Ago's binding siRNAs and miRNAs and Piwi's binding piRNAs. siRNAs are 20–23-nt products of Dicer cleavage, usually from a long >100-nt dsRNA molecule. Although important to antiviral response, many endogenous siRNA (endo-siRNAs) species derived from hairpin RNAs (hpRNAs) or cis-NATs can be found in arthropod genomes (Fagegaltier et al, 2009; Lau et al, 2009; Claycomb, 2014).

---

[1]School of Plant Sciences, University of Arizona, Tucson, AZ, USA   [2]Cellular and Molecular Biology, University of Southern Mississippi, Hattiesburg, MS, USA

Correspondence: alex.flynt@usm.edu

In comparison, piRNAs are Dicer independent and produced by several methods that include endonucleolytic "slicer" activity present in Piwi proteins (Yamaguchi et al, 2020). They are typically 26–30-nt long and have a "U" residue at their 5′ end. In *Drosophila*, two biogenesis modes of piRNAs have been observed that are mediated by three Piwi proteins: PIWI, aubergine (Aub), and argonaute 3 (Ago3) (Ozata et al, 2019). The "ping-pong" mode involves alternating target RNA "slicing" by Piwi proteins. This is an amplifying mechanism where processed RNAs are recruited as new piRNAs. Aub cleaves RNAs that load into Ago3 as secondary piRNAs. Reciprocally, Ago3 substrates load into Aub. The other mechanism relies on piwi proteins cleaving designated transcripts that then become substrates for the RNase Zucchini (Zuc), which they subsequently load into PIWI/Aub in *Drosophila* (Gainetdinov et al, 2018). In *Drosophila*, piRNAs are associated with germline; however, in many other arthropods, piRNAs are found in soma, including hemipterans (Lewis et al, 2018).

Relative to *Drosophila*, whitefly has additional Ago/Piwi proteins and exhibits somatic expression of Piwis. Using small RNA sequencing datasets, miRNAs, endo-siRNAs, and piRNAs were annotated. Many of the loci resembled those described in *Drosophila*; however, we found many instances where there was production of both piRNAs and siRNAs. Using characteristics of these endogenous loci, we designed RNAi triggers that exploit piRNA pathways. Gene silencing triggered by the piRNA pathway were equally as efficient as the siRNA-mediated silencing of endogenous genes. Somatic piRNAs are widespread among insects; however, their application as a pest control tool is yet to be developed. Other hemipteran insect pests and vectors of pathogens to plants and humans, such as pea aphid and the kissing bug, respectively, have been found to produce somatic piRNAs (Brito et al, 2018; Lewis et al, 2018). Insights of this study are expected to apply directly to these pests as well as many others.

# Results

### Whitefly RNAi pathways

Because of the divergent nature of siRNA and piRNA biology, species-specific design is necessary to fully exploit these pathways for effective gene silencing. To characterize the RNAi pathways of whitefly, we first sought to identify the collection of Ago/Piwi proteins encoded in the whitefly "B biotype" (also known as MEAM1) genome (MEAM1v1.2) using existing annotations and BLAST to curate sequences (Fig 1A) (Chen et al, 2016). These sequences

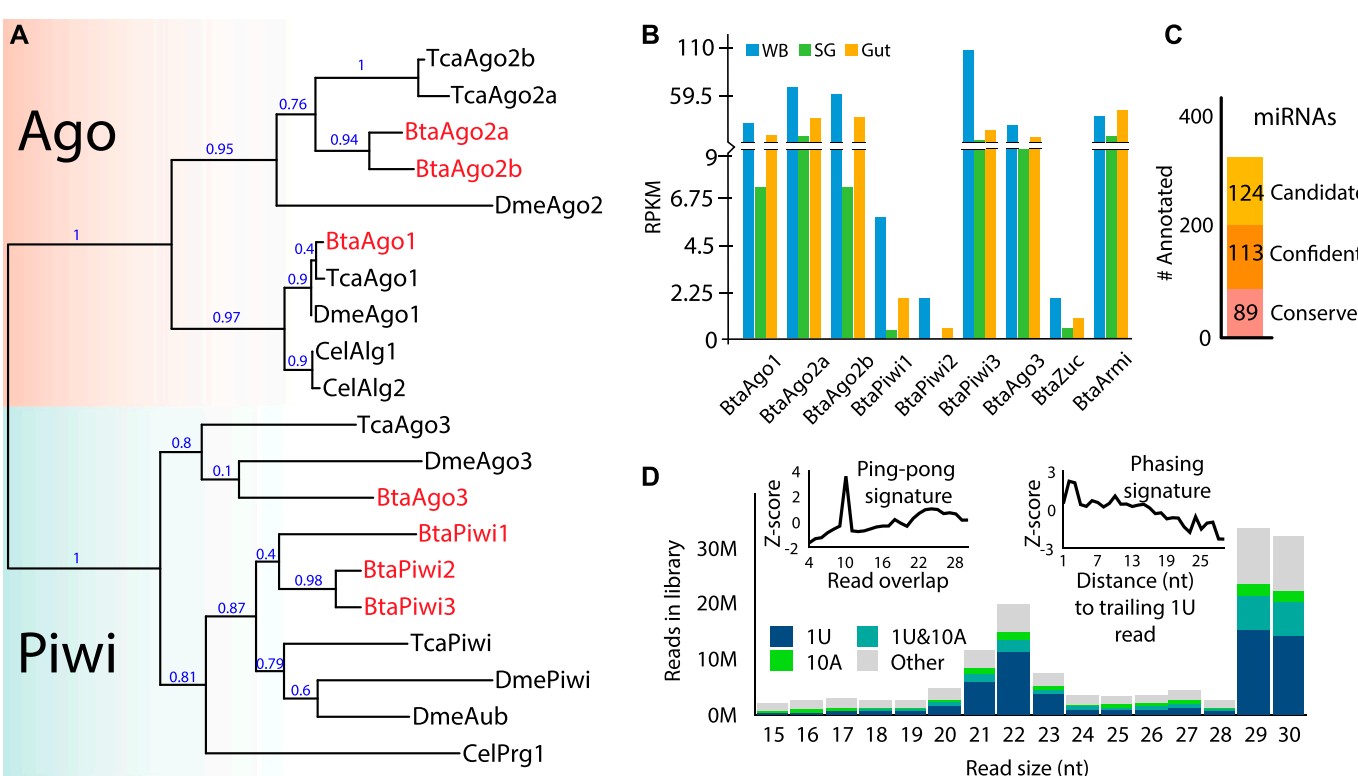

**Figure 1.  RNAi pathways in whitefly (*B. tabaci*).**
**(A)** Relatedness of argonaute and Piwi (Ago/Piwi) proteins from whitefly (Bta) to orthologs in *Drosophila* (Dme), *Tribolium* (Tca), and select family members from *C. elegans* (Cel). Ago and Piwi clades highlighted by colored boxes and whitefly genes in red text. Phylogenetic tree was constructed using the maximum likelihood method. Branch support values shown at nodes. **(B)** Expression determined by RPKM of whitefly Ago/Piwi proteins in whole body, gut, and salivary gland. **(C)** Numbers of miRNAs annotated in this study. Loci are categorized into those conserved with *Drosophila*, novel highly confident, and lower confidence candidates. **(D)** Distribution of small RNA read sizes mapping to the whitefly genome (MEAM v1.2) and piRNA biogenesis modes. Left inset shows read overlap Z-scores to demonstrate the ping-pong piRNA signature of 10-nt overlaps, and right panel distance to trailing 1U reads showing the phasing signature. Bars in the size distribution are colored based on the portion of reads with the sequence identity indicated in the inset legend.

were then compared with Ago/Piwi proteins from *Drosophila melanogaster*, *Tribolium castaneum* (red flour beetle), and subset from *Caenorhabditis elegans*. We found seven members of this family encoded in the whitefly genome. Three of the genes belong to the Ago family with one clearly related to miRNA-loading factors. The two other Agos group with the siRNA-associated Agos appear to be a clade specific duplication as they are not homologs of the two si-Agos seen in *T. castaneum*. This is consistent with the diverging nature of siRNA biology and opens up the possibility that si-Ago function in whitefly might be distinct from fruit flies and beetles. The other four members belong to the Piwi clade. One of the whitefly Piwi proteins is a homolog of DmeAgo3, whereas the other three groups with DmePIWI/DmeAub. This indicates that the ping-pong biogenesis is likely present in these animals. The phasing piRNA pathway is also presumably operative with apparent homologs of Zuc (Bta02312) and the RNA helicase Armitage (Bta07189) (Ishizu et al, 2019).

piRNAs are found in somatic tissues of many insect orders, including hemipterans (Huang et al, 2017; Lewis et al, 2018). To verify if this is also pertinent for whitefly, we investigated RNAi factor expression in the whitefly guts, salivary glands, and whole body (Fig 1B) (Cicero & Brown, 2011). PolyA sequencing libraries from extirpated whitefly guts, salivary glands, and whole body were mapped to the RNAi factor sequences from MEAM1v1.2, per above (Chen et al, 2016). The alignments were then used to calculate Reads Per Kilobase of transcript, per Million mapped reads (RPKM) values for each transcript. Expression of BtaAgo1 and BtaAgo2 was found in all tissues, as well as two Piwi's (BtaPiwi3 and BtaAgo3), along with BtaZuc and BtaArmi (Fig 1B). This contrasts with BtaPiwi2, which is enriched in whole body presumably because of the inclusion of RNAs originating from gonad tissues. This suggests that similar to other hemipterans, whiteflies have somatic piRNAs with both ping-pong and phasing piRNA biogenesis modes being present. Significantly, somatic piRNAs are likely present in whitefly gut, the tissue that would be the primary target of ingested RNAi trigger molecules.

To further investigate whitefly RNAi pathways, endogenous small RNA populations from whole body mixed adults (male and female) were examined using small RNA sequencing libraries mapped against MEAM1v1.2. From this alignment, we first annotated miRNAs using miRDeep2 (Friedlander et al, 2012). Subtracting miRNA-derived reads from datasets would allow focus on non-miRNA small RNA loci such as endo-siRNAs and piRNAs, which unlike miRNAs might have whitefly specific biology. 202 miRNAs are identified with high confidence with 89 being conserved in Drosophila (Fig 1C and Table S1). We also identified 124 additional miRNAs which were classified as lower confidence miRNA candidates because of suboptimal features such as low expression or imprecise precursor cleavage patterns. The miRNA repertoire of the whitefly genome is similar in size to other insects (Kozomara et al, 2019). These results expand on prior miRNA annotations in whitefly because of the increased depth of datasets featured in this study.

Next, we examined the size distribution of reads and found a bimodal read size distribution with peaks at 22 nt representing Dicer products (siRNAs and miRNAs) and 29–30 nt (piRNAs) (Fig 1D). Among the Dicer products, roughly 56% derive from miRNAs. This shows piRNAs are more abundant relative to siRNAs in whitefly. To examine the modes of piRNA production, we analyzed the abundance of read overlap pairs and the distance to 1U trailing reads (Fig 1D). During ping-pong biogenesis, piRNA pairs are cleaved at the 10th base of the guiding RNA. Thus, when this mode is active, piRNAs are found to overlap by 10 bases, which is clear in the dataset. Phasing piRNAs are biased to occur end-to-end and can be recognized by close proximity of trailing reads. Phasing is also evident in the alignment. The abundance of piRNAs is further reflected by the high-proportion 1U reads in the size distribution. Simultaneously, a significant proportion of the reads also exhibit an "A" at the 10th base which would be found on ping-pong pair reads because of pairing with 1U.

## Non-miRNA, small RNA-producing loci in whitefly

Using reads subtracted of miRNAs, we annotated non-miRNA, small RNA-producing loci. 3,873 regions were identified with a read depth greater than 40 and 500+ bp length (Fig 2A and Table S2). The ratio of the number of small (19–23-nt) to long reads (25–30-nt) was then calculated to distinguish whether the locus produced smaller siRNAs (19–23-nt) or longer piRNAs (25–30-nt). This showed the majority of loci appear to be piRNA generating. Only 50 loci had a ratio of small to long that was greater than "one." Interestingly, the piRNA loci spanned regions ranging in size from 500 nt to 50 kb, indicating diverse transcripts generate this small RNA class (Yamanaka et al, 2014). Apparent siRNA-producing loci tended to be shorter regions, of which the longest was about 4 kb in size. We then examined the distribution of read sizes at each locus (Fig 2B). Accumulation was most clear in the piRNA range, which was substantially less for siRNAs. However, a minor signature of siRNA-sized reads could be seen at many loci. To confirm that most loci were sites of piRNA production, we examined read overlaps and trailing 1U read distance, which shows evidence of ping-pong due to 10-nt overlap bias and phasing with juxtaposed trailing 1U reads, respectively. The exception was ~100 loci that showed a greater accumulation of siRNAs.

To verify if these loci are sources of Dicer-produced siRNAs, we sought the 2-nt overhang signature of RNase III processing (Fig 2C). Overlapping read pairs between 15 and 31 nt with this signature were quantified. Pairs were identified where one strand (query read) of a potential duplex overlapped by less than two of its entire length, which would occur with a 2-nt overhang (Antoniewski, 2014). All potential combination of query and complementary target reads were quantified, revealing that 22-nt reads show the greatest evidence of Dicer processing and that this is likely the size of Dicer-2 products. The abundance of apparent Dicer overlapping reads differed from the distribution of the reads in different size ranges, validating this method of characterizing biogenesis. Interestingly, some signal could also be seen in the 29–30-nt sizes that likewise were not reflected in the all read size distribution. This suggests a potential interaction between siRNAs and piRNAs unlike what is reported in *Drosophila* and is consistent with the frequent co-occurrence of siRNA-sized and piRNA-sized reads across all annotated loci (Fig 2B).

Next, we focused on the filter loci by expression to focus on the top 50 long read biased or short read biased loci (Fig S1 and Table S1). The size distribution of reads for each locus was determined, which showed 28–30-nt reads at long read loci, consistent with production of piRNAs (Fig S1). This contrasts with the short read loci, which show signal at 22 and 29 and 30-nts, consistent with co-occurrence of piRNAs and siRNAs seen across all loci (Fig 2B). We also characterized the two groups of loci by length, expression, and

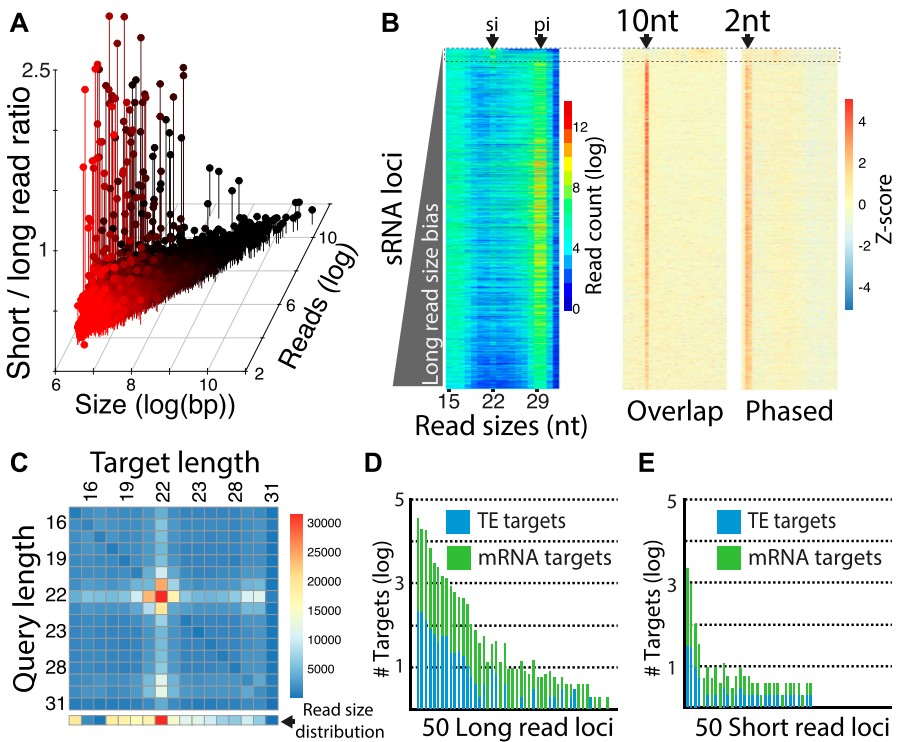

**Figure 2. Whitefly small RNA expressing loci.**
**(A)** Comparison of 3,878 small RNA loci is annotated in whitefly by locus size, number of mapped reads, and the ratio of short (19–23 nt) to long (25–30 nt) mapping reads. **(B)** Visualization of small RNA sizes and piRNA biogenesis signatures for all 3,878 loci. Each row of the heat map represents a locus, which is arranged by read size bias with short read bias at the top and long bias at the bottom. Left panel shows size distribution. Nucleotide sizes are indicated below. Arrows at top show sizes expected to represent siRNAs (si) and piRNAs (pi). Middle panel shows read overlaps quantified by Z-score, arrow shows the 10-nt overlap size. Right panel shows distance of trailing 1U reads; arrow shows the 2-nt proximal read distance. Dashed line box highlights the ~100 loci that do not have piRNA signatures in terms of read size, overlaps, or phasing. This group of loci have more reads at the 22 nt (siRNA) size. **(B, C)** Matrix of Dicer-2 nt overhang signature calculated for loci in the dashed box in panel (B). Read pairs where the query read overlapped by 2 minus its total length were quantified and plotted in the heat map. Line of boxes below the matrix show the read size distribution for reads mapping to the analyzed loci (dashed box in part B). **(D, E)** Number of mRNA and transposable element targets for the 50 most high expressing (Fig S1) (D) loci biased to long reads or (E) loci biased toward short reads.

1U bias (Fig S1). This showed that long read loci are larger, have a greater bias toward 1U, and greater expression, which are characteristics of piRNA clusters. Examining strand mapping showed that high expressing long read loci exhibit bias toward small RNA production from one strand indicating likely single-stranded precursor transcripts converted into phasing piRNAs (Fig S1) (Gainetdinov et al, 2018). The short read loci are predominantly dual-stranded, which is suggestive of a dsRNA precursor serving as a substrate for Dicer (Claycomb, 2014). We also identified 22 hpRNA loci indicating that this variety of locus is present as a minority of the overall collection of whitefly siRNA-generating loci (Fig S2).

To predict the function of these 100 loci, reads aligning to these loci were mapped back to the whitefly genome permitting up to three mismatches. This alignment was then intersected to MEAM1v1.2 annotations (Fig 2D and E). The number of intersections was determined for each locus keeping mRNAs and TEs separate. Both long and short reads target mRNAs and TEs indicating possible roles for piRNAs and siRNAs not only in genome surveillance but also in gene regulatory networks. This is consistent with a proposed role for piRNAs in regulation of protein coding gene expression (Shamimuzzaman et al, 2019). Taken together, this suggests that whitefly siRNAs and piRNAs are gene regulatory factors alongside miRNAs. These observations further reinforce the potential for exploiting these pathways for genetic technology that silences genic transcripts.

### Whitefly endo-siRNA loci are also sources of piRNAs

Prior work in whitefly has shown effectiveness of long dsRNA in gene silencing (de Paula et al, 2015; Malik et al, 2016; Luo et al, 2017; Vyas et al, 2017; Grover et al, 2019). The presumption is that these molecules are processed by Dicer into siRNAs. To better understand small RNAs simulated by fed dsRNA, we used the computational approach described above that finds the 2-nt overhang signature of RNase III cleavage in 20–23 nt reads. Based on this, 76 loci exhibiting apparent Dicer processing were annotated (Fig 3A). When intersecting these Dicer loci with the high expressing long and short read loci (Fig S1), 42 short read loci and only one long read locus have the Dicer processing signature (Fig 3A). Seqlogo analysis was also performed on the 22-nt Dicer reads showing a bias for 1U and a matched 20th base A (20A). Among the other bases, 1G residues were disfavored along with the paired 20C.

Next, we inspected individual loci to understand their function and biogenesis. The Dicer locus that overlapped with the one long read locus is an interesting genomic site (Fig 3B). This region is a large phasing piRNA precursor with an annotated, interior antisense transcript. The Dicer signature reads coincide with this antisense transcript that seems to form a dsRNA with the piRNA precursor. Other Dicer loci also arise from overlapping antisense transcripts. Indeed, many cis-NAT siRNAs are observed in the Dicer annotations, with one such example shown from Scaffold1098 (Fig 3B).

Through curation of the annotations, loci were placed in five categories: siRNA, cis-NAT, No bias, piRNA, and piRNA cluster (Fig 3C and Table S1). These groupings were determined by evaluating dominant small RNA size and the dominant processing signatures of read pairs—2-nt overhangs for Dicer or 10-nt overlaps for ping-pong piRNA. The siRNA group is located in intergenic regions and has a strong bias toward short reads that appear to be dicer processed. The cis-NATs were sites of siRNA production between opposing mRNAs as showing in Fig 3B. In addition to these predominantly

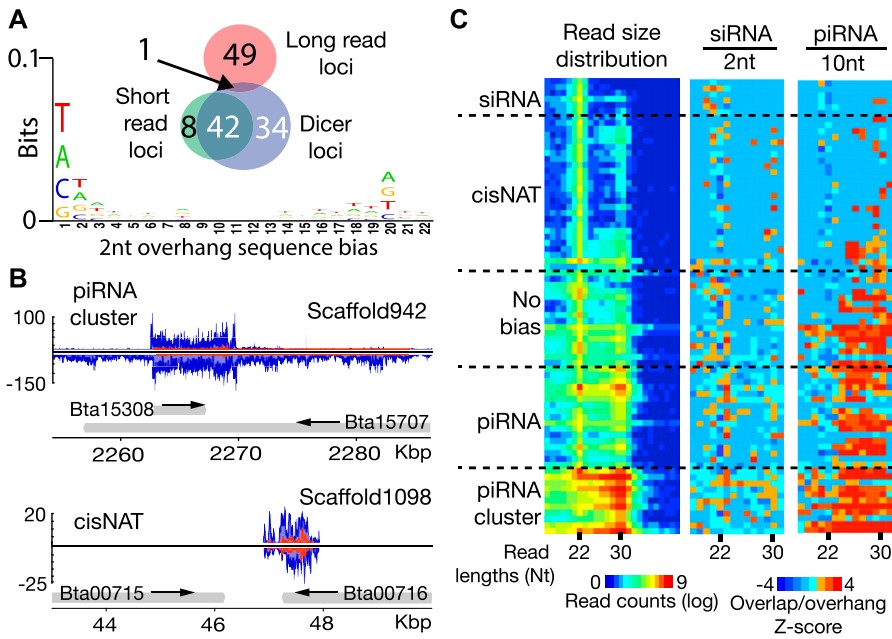

**Figure 3. Characterization of whitefly loci with Dicer cleavage signature.**
**(A)** Intersection of Dicer processing loci showing 2-nt overhangs for reads sized 20–23 nt with long and short read loci. The sequence biases of Dicer read loci are shown below in the seqlogo graphic. **(B)** Appearance of Dicer produced small RNAs (siRNAs) at sites of convergent transcription. Top panel shows expression of siRNAs in a piRNA cluster. Bottom panel is a cis-natural antisense transcript (cis-NAT). Blue trace shows all reads mapping to locus. Read trace shows reads with Dicer-2-nt overhang cleavage pattern. **(C)** Read size distribution and biogenesis pattern of small RNAs produced at 76 Dicer signature loci. Length of reads in heat maps is indicated below. Curated identities are shown on the left. The leftmost heat map shows the distribution of reads sizes, middle shows z-scores for 2-nt overhangs (siRNAs), and right heat maps show z-scores for 10-nt overlaps (piRNAs).

siRNA-producing regions that are similar to ones observed in *Drosophila* (Czech et al, 2008; Ghildiyal et al, 2008), many loci produce both siRNAs and piRNAs. These dual-identity loci could be grouped into one of three categories. One where there was equal production of siRNAs and piRNAs (No bias), a second for which some siRNAs were present, but piRNAs are dominant (piRNA), and the third group that harbors large piRNA clusters with only a minor production of siRNAs, the latter being similar to the locus shown in Fig 3B. These observations show that despite an Ago repertoire similar to *Drosophila*, small RNA biogenesis in whitefly is distinct. This provides an opportunity to exploit these divergent activities for gene silencing and pest management.

### Metabolism of exogenous dsRNA by whitefly

We then extended our evaluation of processing dsRNA transcripts to those introduced exogenously via feeding. Here, we tested three off-target, synthetic dsRNAs dissolved in a sucrose solution fed through an artificial system. The RNAs cloned from genes of the potato psyllid *Bactericera cockerelli* (Sulc.) were fed to adult whiteflies from which small RNA and messenger RNA sequencing libraries were generated. Significant accumulation of reads arose exclusively from dsRNAs and not from other sections of the psyllid gene from which they were cloned (Fig 4A). However, only a fraction of the reads show a signature of Dicer processing based on 2-nt overhangs. This suggests that most of the synthetic RNA was likely degraded with only a minority entering the siRNA pathway, which is reflected in the distribution of reads produced from the dsRNA sequences (Fig 4B). Only a modest peak was seen at 22 nt with many more at the 15 nt size. The low efficiency is likely in part caused by dsRNA-specific nucleases (dsRNases), which are common in hemipteran insects. Several dsRNases from gut and other tissues have been identified in whiteflies (Luo et al, 2017; Singh et al, 2017).

Furthermore, the alkaline pH of sternorrhynchan midgut is also compromised by RNA integrity (Cristofoletti et al, 2003; Molki et al, 2019).

Using these datasets, we sought to identify similarities between small RNAs derived from fed dsRNA and endogenously expressed siRNAs. Specifically reads were subsetted based on the sequence content to find population where signatures of dicer processing were most evident (Fig 4C). This was guided by the seqlogo results of endogenous siRNAs that showed preference for 1U and depletion of 1G (Fig 3A). In unfiltered reads, only a slight enrichment of 22-nt RNAs was seen with no evidence of 2-nt overhangs. Next, the reads were extracted based on their 5′ residue, which showed similar size distribution to the unfiltered library with the exception of 1G reads where there was no bias toward 22-nt reads. However, for each subset, no 2′ overhang specific to 22-nt reads was observed. The analysis was then extended to include not only the first base of the read but also the 20[th] base. When considering 1U/A/C(H) and 20A/U/G(D), greater abundance of 22-nt reads was seen but still no substantial 2-nt overhang Dicer signature. When 1U/A(W) and 20A/U(W) were examined, an even greater enrichment of 22-nt reads as well as 2-nt overhang enrichment for this size was observed. For individual nucleotide pairs (1U-20A, 1A-20U, 1C-20G, and 1G-20C), read size and overhang enrichment increased, particularly for 1U-20A in size distribution and 1A-20U for 2-nt overhang. This analysis provides a framework for computationally isolating siRNA processing signatures from degradation products, which is essential when considering exogenous dsRNA processing because of cloning of digestive contaminants.

To understand the physiological consequences of ingesting dsRNA, we examined the effect on expression of the small RNA loci annotated in this study and protein-coding genes. Studies in other animals suggest there may be competition between exogenous and endogenous small RNAs for biogenesis pathways and that dsRNA

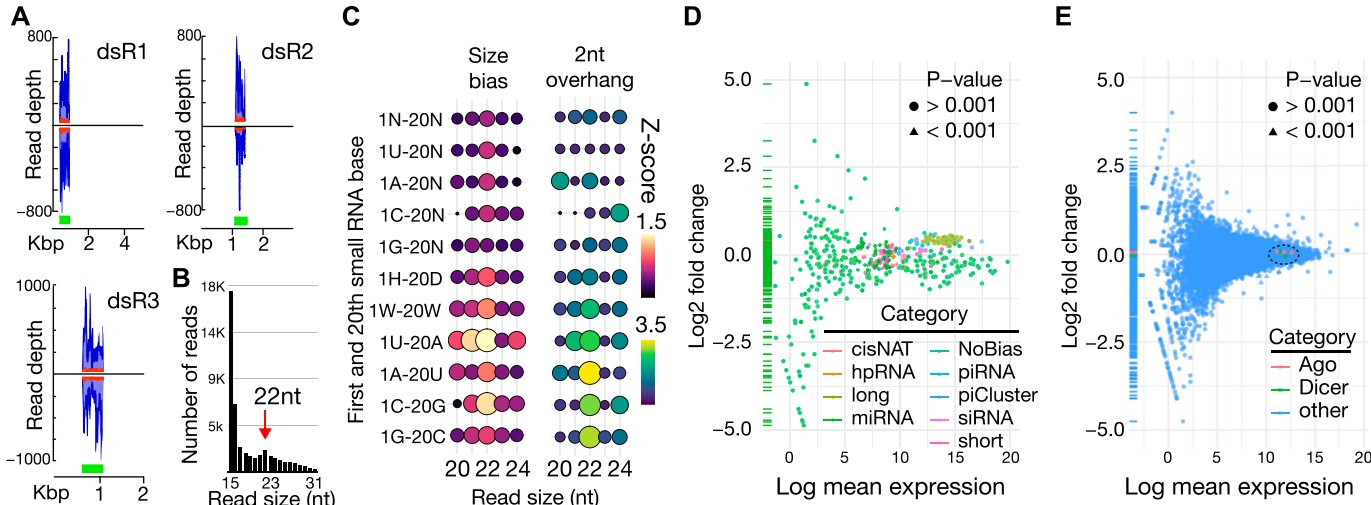

**Figure 4. Metabolism of exogenous dsRNAs in whitefly.**
**(A)** Accumulation of reads mapping to dsRNA sites (green boxes) in the context of the originating transcript from potato psyllid (*B. cockerelli*). Blue trace shows all reads mapping to the locus. Read trace shows reads with Dicer cleavage pattern. **(A, B)** Size distribution of read derived from the three off-target dsRNAs (shown in (A)). Red arrow shows the expected size of siRNAs (22 nt). **(C)** Balloon plot showing characterization of sequence biases in exogenous siRNAs. Read sizes are indicated below. Color and diameter of circle scale with Z-scores quantifying different size reads. On left, the sequence identities of small RNA subsets are indicated for the first base of the read and the 20th base of the read. N = any residue, H = U/A/C, D = A/U/G, and W = A/U. The left group of balloons show the abundance of reads, and the right group of circles abundance of reads with 2-nt overhangs. **(D, E)** Differential expression of (D) small RNA loci and (E) mRNAs between whiteflies treated with water or the three off-target dsRNAs. Data points colored by identity. Circles represent nonsignificant change in expression, triangles significant. Dashed circle shows location of Dicer and Ago proteins in the scatterplot.

may be recognized as a viral motif resulting in the activation of defense pathways (Jelinek et al, 2011). After feeding dsRNA, small RNA sequencing showed no significant change in endogenous small RNA expression compared with control (Fig 4D) (Table S3). For protein coding genes, we observed about 500 transcripts that were differentially expressed based on a *P*-value ≤ 0.001 (Fig 4E and Table S3). Only 20 of these genes exhibited a log(fold2) value greater than 2 or less than –2. All genes in this group have very low expression with 14 having unknown function. The genes having a known identity appear to be involved in basic metabolism or development. The one exception is an RNase H-containing gene (Bta15726) that could be involved in an antiviral response. However, it is down-regulated, which is inconsistent with being deployed to combat perceived viral infection. Thus, it would seem that whitefly does not mount an antiviral-type response to dsRNA. We also observed no change in expression of RNAi factors such as the Ago and Dicer proteins, suggesting that whitefly can metabolize exogenous RNAi triggers without affecting its core RNAi processes. Taken together, it appears that when ingested, the bulk of dsRNA is degraded with a small amount contributing to the siRNA pool and that exposure to dsRNA has a minimal impact on off-target gene expression in whitefly.

### Exploiting somatic piRNAs in addition to siRNAs for gene silencing

In this study, we found a significant population of piRNAs, which are more abundant than the endogenous siRNAs—the species exploited by existing RNAi approaches. The piRNAs also appear to be expressed in soma and show potential widespread control of mRNAs and not just a role in genome surveillance. This suggests

that the piRNA pathway might be exploited to silence endogenous gene expression in whiteflies as an alternative method to the classic dsRNA-based siRNA strategy.

To trigger ectopic production, we engineered recombinant nucleic acids that take advantage of the major principle of piRNA biogenesis—recruitment of Piwi-cleaved fragments into the pathway (Fig S3). We fused sequences from two loci annotated in this study, a piRNA bias locus (piRB-6) and siRNA–piRNA no bias locus (No bias-14) to target gene sequences. Both loci were among those that showed evidence of Dicer processing as well as piRNA production (Fig 3 and Table S1). We chose two different genes to target with these constructs: aquaporin1 (*AQP1*) and alpha glucosidase1 (*AGLU1*), which were used in a previous study that yielded high gene knockdown via dsRNA (Vyas et al, 2017). To explore design principles, the positive strand of the locus was fused to *AQP1* and the negative strand to *AGLU1*.

Using these constructs, both synthetic dsRNAs and ssRNAs (single-stranded RNA) were generated and fed to whiteflies in the artificial system described above. The concentration of RNAs (30 ng/μl) used was similar to what was previously fed to whitefly in dsRNA experiments (Vyas et al, 2017). Luciferase sequences fused to piRB-6/No_bias-14 were used as off-target controls. After feeding access for 6 d, expression of target genes was assessed by qRT-PCR (Fig 5A). As previously reported, dsRNAs elicited gene knockdown of 68–80%. Satisfyingly, the piRNA triggers showed a similar degree of gene silencing with reduction in target expression of 60–80%. This result was observed for both ssRNA and dsRNA triggers with both piRNA sequences (piRB-6, No_bias-14) and targets (*AQP1*, *AGLU1*). For *AQP1*, piRNA triggers were equal to dsRNA (conventional dsRNA), whereas *AGLU1* was not as well down-regulated by the piRNA

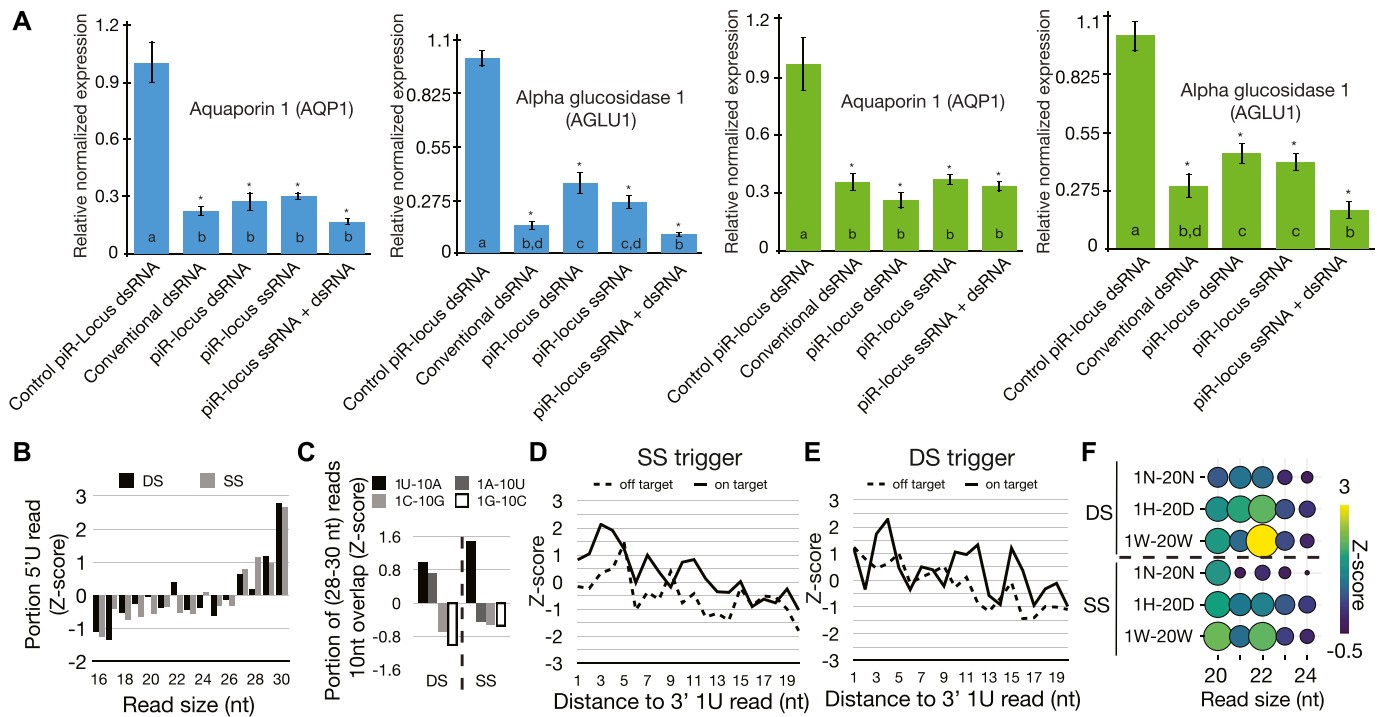

**Figure 5. Exogenous piRNA-mediated gene silencing in whitefly.**
**(A)** Relative expression of *AQP1* and *AGLU1* genes determined by qRT-PCR after feeding with synthetic RNAs generated from piRNA triggers. Blue bar graphs are results when target gene sequences are fused to sequence from piRNA-biased locus 6 (piRB-6) sequences. Green graphs are when they are fused to No Bias-14 sequences. At least three independent biological replicates were used for each type of feeding. Error bars show standard error, and letters indicate significance groups determined by Tukey's HSD test. *$P \leq 0.05$. **(B, C, D, E, F)** Analysis of small RNA-sequencing data from animals fed piRB-6–based piRNA triggers that map to the synthetic RNAs. **(B)** Portion of small RNA-sequencing reads with 1U residues shows biased to long (piRNA) sized reads. Black bars are from double-stranded (DS) triggers and gray from single-stranded (SS) versions. **(C)** Enrichment of ping-pong piRNA pairs in longer sized RNAs (28–30 nt) in the target gene region of the piRNA triggers. Sequence identities are indicated in the legend. DS, double-stranded triggers; SS, single-stranded triggers. 1U-10A reads, which are characteristic of bona fide ping-pong piRNAs show the greatest abundance. **(D, E)** Phasing signature plots separated by off-target and on-target strands for (D) single-stranded piRNA triggers and (E) double-stranded triggers. **(F)** Balloon plot showing reads with Dicer-2 nt overhangs for the DS and SS triggers. Color and size of circles scale with the abundance of 2-nt overhang pairs. Left shows the sequence identities of small RNAs analyzed (N = any residue, H = U/A/C, D = A/T/G, and W = A/T).

triggers relative to dsRNA, suggesting inclusion of positive strand sequence might lead to superior knockdown. However, by combining ssRNA and dsRNA piRNA triggers for either sequence, gene silencing became comparable to conventional dsRNA for *AGLU1*. These results provide robust evidence that piRNA triggers, even those that comprised ssRNA, are capable of gene silencing in organisms that share RNAi biology with whiteflies.

Small RNAs were then sequenced to characterize the processing of the piRNA triggers. Small RNAs were sequenced from animals fed piRB-6 dsRNAs and ssRNAs targeted to both *AQP1* and *AGLU1* (Figs 5B–F and 6). Reads mapping to these triggers showed significant heterogeneity in read size with no accumulation of a specific size, indicating the bulk of fed RNAs were degraded. To identify potential small RNAs among the detritus, we determined the relative abundance of 1U reads, a characteristic of piRNAs as well as siRNAs (Fig 5B). From this, we found significantly more 26–30-nt piRNA-sized reads. In the double-stranded treatment, a small peak possibly corresponding to 22-nt siRNAs could be observed, but not for the single-stranded piRNA triggers.

Next, we focused on the identity of small RNAs produced against the target gene. piRNA biogenesis could be observed for both triggers but more so for the single-stranded versions (Fig 5C). Ping-

pong processing was observable when comparing the number of overlaps for different nucleotide combinations that can form pairs: 1U/10A, 1A/10U, 1C/10G, and 1G/10C. Read pairs were determined for all ranges of reads, and for those in piRNA sizes (28–30 nt). The greatest enrichment for 28–30-nt reads was seen for those with the signature of ping-pong piRNAs: 1U/10A. Phasing was also assessed for each strand of the piRNA triggers (Fig 5D and E). This biogenesis mechanism was evident for the transcribed strand of single-stranded triggers, which is complementary to the target genes (*AQP1* and *AGLU1*). For both strands of the double-stranded trigger and the potential target-derived reads in the single-stranded fed condition, less phasing was evident, although a noticeable trend toward close proximity of 1U reads was seen. Next, we investigated whether siRNAs were processed from the triggers by examining 2-nt overhangs in read populations as in Fig 4C. When reads with 1U/A/C-20U/A/G or 1U/A-20A/U were examined, the double-stranded trigger showed a greater number of 22-nt Dicer signature reads (Fig 5F). Together, these results show that regardless of whether the trigger is double-stranded or single-stranded, piRNAs are produced. However, there is less piRNA production from dsRNAs. Presumably, accessing the piRNA pathway requires an unwinding step for dsRNAs mediated by gut or cellular helicases, reducing the

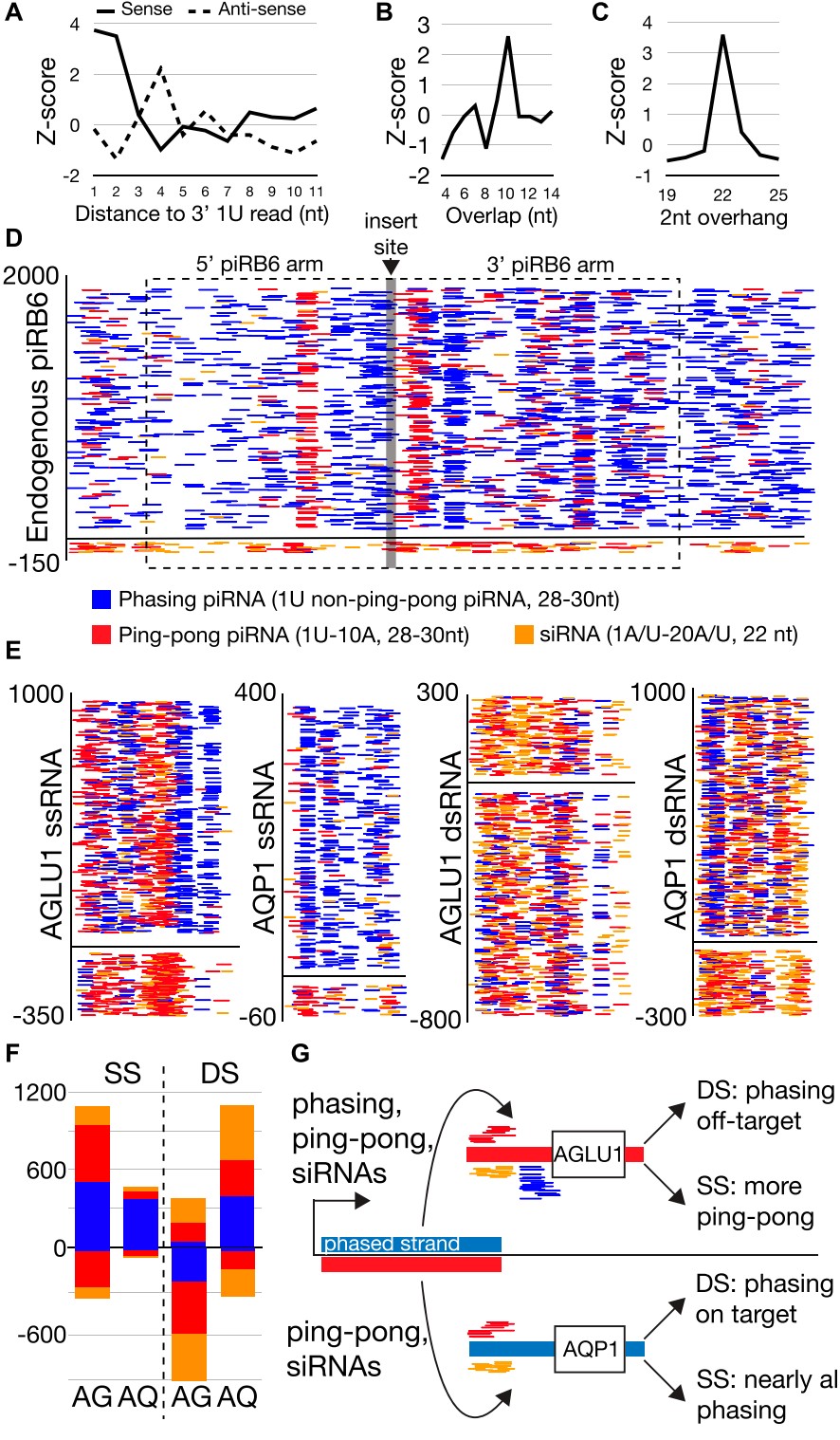

**Figure 6. Processing of piRNA triggers.**
**(A, B, C, D)** Characterization of the piRB-6 locus.
**(A)** Phasing analysis of trailing 1U reads shows greater phasing signature on the plus strand of the locus compared to the antisense strand. **(B)** Overlap analysis for the piRB-6 locus showing a peak at 10-nt overlaps. **(C)** Enrichment of 22-nt reads that overlap by 2 nt at piRB-6. **(D)** Read accumulation at piRB-6. Alignments are colored by identity. Blue represents phasing piRNAs characterized by long 28–20-nt 1U reads that do not overlap by 10 with antisense reads and therefore unlikely to be involved in ping-pong. Red are ping-pong piRNAs being 28–30-nt reads that have 1U/10A sequences that also overlap by 10. Orange is siRNAs being 22-nt reads that have 2-nt overhangs with a 1U/A and 10A/U. The region cloned for the piRNA triggers indicated by dashed line box. The site of target sequence insertion is shown by the gray line. Y-axis shows read density. **(E)** Read accumulation using the color coding in part D at the sequence target region of piRNA triggers. Similarly, y-axis represents read density. Positive strand depicted at top of graphs is complementary to target. **(E, F)** Quantification of read identities by strand for plots shown in part (E). **(D, E)** Color scheme same as used in (D, E). AG = AGLU1 and AQ = AQP1. **(G)** Diagram showing the consequences of using different piRNA trigger configuration. Blue represents phasing strand of piRB-6 and red the complementary. **(D)** Same color scheme in (D) used to represent reads.

entry of double-stranded triggers. In comparison, the double-stranded triggers give rise to more production of siRNAs.

To understand the differences in target knockdown by the different piRNA trigger configurations, we investigated the biogenesis of small RNAs from each. Before examining the exogenous triggers, we investigated more deeply small RNA production from the endogenous piRB-6 locus used to make the piRNA triggers (Fig 6A–D). This region shows clear piRNA phasing, ping-pong, and siRNA biogenesis (Fig 6A–C). Read alignments of each biogenesis mode were visualized at the locus (Fig 6D). 28–30-nt 1U-10A reads

overlapping by 10 nt represent ping-pong reads. Phasing piRNAs are reads 28–30-nt long with a 1U that did not show a 10-nt overlap. siRNAs are reads that start with 1U/1A and a 20A/20U also showing a 2-nt overhang. At this locus, the positive strand of the locus shows nearly 20-fold accumulation of small RNAs. This is clearly due to phasing piRNAs on the positive strand, and only modest accumulation of ping-pong piRNAs and siRNAs on the negative strand.

The asymmetry of read expression at the piRB-6 locus appears to cause the difference in gene silencing for the two configurations of piRNA triggers (Fig 5A). This is apparent when the accumulations of small RNA types are examined for the gene targeted region of each trigger (Fig 6E–G). As with the endogenous loci, we quantified 1U-10A ping-pong piRNAs, 1U non–ping-pong piRNAs (phasing), and siRNAs with 1U/A-20A/U. The *AGLU1* trigger is composed of the antisense of piRB-6 and could be targeted by sense-phasing piRNAs, ping-pong piRNAs, and siRNAs. For the single-stranded version of the *AGLU1* trigger, we observe significant accumulation of ping-pong piRNAs and phasing on the strand synthesized and fed. The ping-pong piRNAs complementary to the trigger are likely derived from the target gene, which is robustly silenced by this trigger. This contrasts with the double-stranded *AGLU1* trigger, which shows that the off-target strand is much more robustly converted into small RNAs, particularly presumptive phased piRNAs. This explains the lower silencing efficiency for double-stranded *AGLU1* trigger (Fig 5A). The off-target strand of the *AGLU1* dsRNA trigger duplex is the strand that is phased in the endogenous locus.

This same phenomenon is seen in *AQP1* triggers which sport the sense strand of piRB-6 for the on-target strand. For single-stranded *AQP1*, nearly all the RNAs appear to be phasing piRNAs, and for the double-stranded version, most of the phasing piRNAs are on-target. Both of these trigger versions lead to robust gene silencing. These results indicate that a superior choice for piRNA trigger design is to select the phased strand of piRNA loci to fuse with gene-targeting sequences. It also shows that the small population of endogenous antisense ping-pong piRNAs or possibly even the siRNAs has a heightened role in promoting phasing. This is an intriguing departure from *Drosophila* where trailing piRNAs are produced downstream of a site of Piwi protein-initiated cleavage. Here, it seems phasing of piRB-6 can be initiated internally because the region cloned for these triggers only includes an interior section of the locus (Fig 6D). It is also clear from these results that double-stranded triggers, expectedly, lead to greater production of siRNAs.

## Discussion

This study provides an in-depth analysis of the RNAi pathways in *B. tabaci*, a hemipteran insect pest and plant virus vector, and offers a rationale design of piRNA-based gene silencing biotechnology. Most significantly, we show ingested RNAs can enter piRNA pathways, which opens up the possibility for an entirely new strategy for gene silencing and potentially commercial products. On a superficial level, whitefly small RNAs seem similar to *Drosophila*. There are three distinct types of small RNAs (miRNAs, siRNAs, and piRNAs), as in fruit flies. However, upon close inspection, the biogenesis and function of the endogenous small RNAs in whitefly are quite

different. Our work reinforces the consistent observation that non-miRNA RNAi pathways are fluid; clade-specific duplication of the RNAi factors is common, even loss of an entire class of small RNA has occurred in several metazoan clades (Sarkies et al, 2015; Calcino et al, 2018; Mondal et al, 2018). Furthermore, these findings illustrate the benefits of in-depth dissection of the RNAi biology for evolutionarily and biologically different organisms, beyond those examined in model study systems, for developing genetic technology.

Through this comprehensive annotation of whitefly small RNA loci, more than 200 novel miRNAs are described, as well as 3,878 siRNA or piRNA loci. Previously described configurations whitefly siRNA and piRNA loci were observed such as large single-stranded, phased piRNA loci and siRNA expressing cis-NAT and hpRNA loci (Figs 3 and S2 and Table S1). However, curation of loci found extensive evidence of siRNA and piRNA biogenesis occurring simultaneously at many loci. In fact, this appeared to be the rule for most endogenous siRNA and piRNA genes, and seemingly, is related to a different biogenesis and function for whitefly siRNAs or piRNAs. In *Drosophila* and vertebrates, piRNAs mainly control TEs in germline; however, many of the piRNA pathway accessory proteins such as Rhino, Deadlock, Cutoff, and Moonshiner from the Drosophilids are not conserved indicating that piRNAs are shaped to individual organism's biology in an evolutionary arm race between the piRNAs and their targets (Ozata et al, 2019). Indeed, it is predicted that abundant somatic piRNAs engage gene regulatory networks in many basal arthropods, such as hemipterans, suggesting that this is the ancestral piRNA biology (Lewis et al, 2018). This combined with observations that whitefly piRNAs respond to viral infection suggest diverse roles for these small RNAs in this insect (Shamimuzzaman et al, 2019). Moreover, we find that phasing biogenesis can be initiated in the interior of loci as only a portion of the pBias-6 locus as used in the synthetic RNAs. This is distinct from the trigger/responder/trailing piRNA arrangement seen for phased *Drosophila* piRNAs where a trigger and responder piRNA interaction initiates phasing. In the whitefly system, it appears that antisense small RNAs, perhaps either siRNAs or piRNAs, may be able to slice the transcript and divert it into phasing type biogenesis. Alternatively, a mechanism may be involved where phasing is not triggered by slicing as seen in *Drosophila* follicle cells (Lau et al, 2009).

Although RNAi has been successful for controlling some pests such as coleopterans (beetles), many other pests such as some lepidopterans (moths and butterflies) are unresponsive to exogenous RNAi trigger (Shukla et al, 2016; Parsons et al, 2018). Penetrance of RNAi in hemipteran insects is moderate, and higher dosage of dsRNA is required (Joga et al, 2016). pH in the gut of the hemipteran insects is basic, and presence of the nucleases in the gut has been reported in whiteflies, aphids, and other hemipteran insects (Luo et al, 2017; Singh et al, 2017). We have noticed in this study that only a minority of the reads produced from the dsRNA trigger are siRNAs (Fig 4A and B). This could be attributed to low abundance of the intact dsRNA for uptake by the gut epithelium cells. We see a similar accumulation of degradation products with piRNA triggers. Interestingly, even with ssRNA triggers, we see significant accumulation of small RNA reads from synthetic RNA along with some antisense reads. The antisense reads we observe have dominant ping-pong and minor siRNA signature. How the ssRNAs trigger production of these molecules is not clear but could

involve the recruitment of target mRNA into small RNA biogenesis. Although this is the standard behavior for piRNAs, it is not typical for siRNAs in organisms that do not possess Rdrp activity (Sarkies et al, 2015; Almeida et al, 2019; Pinzon et al, 2019). We view this result as a first glimpse at a heretofore unappreciated small RNA biogenesis mechanism that involves interaction between siRNA and piRNA biogenesis, consistent with the widespread co-occurrence at endogenous loci. Encountering such an unknown interaction is not entirely surprising as this study represents the first effort to characterize small RNA biogenesis on a per locus level in a non-holometabolous insect. This further reinforces the value of knowledge-based RNAi design gleaned from investigating exogenous trigger processing. In this study, we provide clear rules for maximizing piRNA production, which could be fundamental to potent gene silencing technology aimed at aphids, mealybugs, psyllids, whiteflies, and other hemipterans.

As hemipteran insects respond to exogenous long dsRNA-mediated RNAi trigger only moderately, using the gene silencing function of the piRNA pathway is exciting. These results show that in whitefly, although there is significant sensitivity to dsRNA, there is very little physiological response to dsRNA feeding. Even the secreted gut dsRNases do not become transcriptionally activated by feeding. This will likely apply to other hemipteran herbivores with similar composition of RNAi pathways and dsRNases. We expect that piRNA triggers, single-stranded or double-stranded, will likewise be physiologically neutral. The most promising result we report is that exogenous piRNA triggers are as effective as the siRNA versions. This study provides the first report of the exploitation of piRNAs as a feeding-based insect pest control strategy. Thus, this approach could become key for designing effective RNAi approaches against many insect pests that are found to be resistant to dsRNA-mediated RNAi. Finally, dsRNAs are capable of activating interferon response in humans and other vertebrates through binding of TLR3 receptors (Zhang et al, 2016). Deploying ssRNA piRNA triggers as a pest control approach would avoid activating this pathway. As a result, beneficial, non-pest organisms in the field would also be spared from off-target effects of dsRNAs as piRNA triggers rely on the specific genomic sequence of the target species and would not be converted into siRNAs as that happened with dsRNA-based triggers. Taken together, these findings demonstrate the benefit of in-depth studies of non-model organismal RNAi biology and demonstrate that somatic piRNAs can be used for environmental RNAi.

# Materials and Methods

### Whitefly colony maintenance

Insects in this study came from the type *B. tabaci* Arizona B biotype (AZ-B) whitefly colony established in Brown laboratory in 1988 after its discovery on poinsettia plants in Tucson, Arizona (Vyas et al, 2017). For this study, AZ-B adult whiteflies were serially transferred to and reared on cotton (*Gossypium hirsutum* L. cv Deltapine 5415) plants at the 8–10 leaf stage.

### Phylogenetic tree construction

*T. castaneum* sequences of the argonaute proteins were downloaded from National Center for Biotechnology Information (NCBI) (EFA09197.2, Ago1; EFA11590.1, Ago2a; EFA04626.2, Ago2b; EFA02921.1, Ago3; and EFA07425.1, Piwi). Whitefly sequences were downloaded from *B. tabaci* MEMA1 genome database: ftp://www.whiteflygenomics.org/pub/whitefly and the argonaute sequences were curated using blast and protein domain search tools InterPro and ScanProsite. The final argonaute genes are *Bta01840, BtAgo1; Bta00938, BtAgo2a; Bta12142, BtAgo2b; Bta04637, BtAgo3; Bta00007, BtPiwi1; Bta00198, BtPiwi2; and Bta08949, BtPiwi3*. Annotated *D. melanogaster* and *C. elegans* sequences were also obtained from NCBI. The phylogenetic tree shown in Fig 1A was reconstructed in http://www.phylogeny.fr suite. Multiple sequence alignment was carried out using MUSCLE, phylogenetic tree was constructed by maximum likelihood method, and the maximum likelihood tree was visualized by TreeDyn.

### Cloning of whitefly sequences and in vitro transcription of ssRNA and dsRNA

*AQP1* (KF377800.1) and *AGLU1* (KF377803.1) sequences from a previous study (Vyas et al, 2017) were cloned in pGEMT-easy vector. The cloned plasmids were used as templates for PCRs, which were used in ssRNA and dsRNA synthesis reactions. For creating the fusion constructs (adding piRNA/siRNA sequences to the gene of interest [GOI]: *AQP1, AGLU1,* and Luciferase sequences), the SOEing PCR method was followed (Supplemental Data 1). 238- and 199-nt-long region from No_bias-14 locus (Scaffold40734: 1537-1774, 1811-2009) were fused to the left and right sites of the GOI, respectively. From the piRB-6 locus, the left and right flanking sequences were 342 and 366 nt, respectively (Scaffold185: 15168-15509 and 15616-15981) (Supplemental Data 1). All Six fusion constructs were cloned into pGEMT-easy plasmid for double-stranded and ssRNA synthesis. 231-nt luciferase gene sequence from psiCHECK-2 (Cat. no. C8021; Promega) vector was cloned into the pGEMT-easy vector. ssRNA and dsRNA from the luciferase sequence was used as control RNA.

Each of the piRNA trigger constructs consisted of three parts, which were PCR-amplified from whitefly cDNA using Phire Plant Direct PCR Master Mix (Cat. no. F160S) following the manufacturer's instruction. During these PCRs, 30-nt sequence from the left and right flanking regions were added to the GOI (*AQP1, AGLU1,* and Luciferase) sequences by adding the sequences in the forward and reverse primers of the GOI. Gel-extracted PCR products (GeneJET Gel Extraction Kit, Cat. no. K0691) were then ligated using two separate SOEing PCRs. First, the left flanking sequence was attached to the GOI and gel-extracted. In the second step, the fusion product from the first step was ligated to the right flanking sequence. These sequences are provided in the Supplemental Data 1.

PCR products with T7 promoter sites on both strands were used for dsRNA synthesis, whereas for ssRNA, PCR was carried by allowing the T7 promoter site in one strand. PCR products were directly used to synthesize the synthetic RNAs using MEGAscript T7 Transcription Kit (Cat. no. AM1334; Thermo Fisher Scientific) following the manufacturer's protocol.

## Oral delivery of the synthetic RNAs to whitefly, RNA extraction, and qRT-PCR

Using a hand-held aspirator, 100 adult whiteflies were collected for each biological replicate from the colony and transferred to a plastic feeding chamber. 200 µl of 30 ng/µl RNA in 20% sucrose solution was sandwiched between two sterile Parafilm M layers, and feeding access to the solution was given to the insects for 6 d. On day 6, the insects were collected for RNA extraction.

Total RNA was extracted following the standard TRIzol RNA extraction method. The extracted RNAs were DNase I–treated (DNA-Free kit, Lot 00522653; Invitrogen) and 2 µg RNA was used for cDNA synthesis using High Capacity cDNA Reverse Transcription Kit (Lot 00692533; Applied Biosystems). The TaqMan qPCR master mix (Universal PCR Master Mix, Lot #1908161; Applied Biosystems) was used for quantitative gene expression analysis using standard protocol. Whitefly 18S ribosomal RNA gene was used for normalizing the expression of the target genes. All qRT-PCR primer sequences from a previous study were used in this study and can be found in Supplemental Data 1 (Vyas et al, 2017). Each treatment and control groups of the synthetic RNA feeding were carried out using at least three independent biological replicates. The ΔΔCt method was used for gene knockdown analysis. $t$ test and one-way ANOVA were used for statistical analysis in CFX Maestro software v1.1.

## mRNA library preparation, sequencing, and gene expression

Total RNAs were extracted using conventional the TRIzol RNA extraction method from different manually dissected tissues of whiteflies (gut, salivary gland, and whole body) (Cicero & Brown, 2011). RNA integrity was confirmed using an Agilent 2100 Bioanalyzer (Agilent Technologies Inc.). Sequencing libraries were constructed using Illumina's TruSeq RNA Sample Preparation Kit v2, Cat. no. RS-122-2002 (Set B). Using magnetic oligo (dT) beads, only poly(A) tail containing RNAs were separated from total RNA. Next, the mRNAs were fragmented by zinc treatment, and the first-strand cDNA was synthesized from the fragmented RNAs using SuperScript II reverse transcriptase and random primers from Invitrogen. Then second-strand cDNA was synthesized, and Illumina multiple indexing adapters were ligated to the fragments. The remaining library construction steps were carried out following the manufacturer's protocol. Quality filter and processing of the sequenced reads were performed using Illumina CASAVA v1.7.0, FastQC, and Trimmomatic. For each of the RNAi factors analyzed (Fig 1B), the reads were mapped with Bowtie2 to transcript sequences from the whitefly genome database www.whiteflygenomics.org (Langdon, 2015; Chen et al, 2016). bedtools was used to count read alignments to each transcript (Quinlan & Hall, 2010).

## Small RNA library preparation and sequencing

Total RNA was extracted from adult whiteflies using the standard TRIzol RNA extraction protocol. After the DNase treatment, small RNA-seq libraries were constructed using NEXTflex Small RNA-Seq Kit v3 (NOVA-5132-06). First, A 3′ 4N adenylated adapter was ligated to the 3′ end and 5′ standard Illumina adapter was ligated to the 5′ end of the RNAs. Reverse transcription was carried out on the adapter ligated RNAs. Synthesized cDNAs were PCR-amplified, and each sample was barcoded with I7 Illumina-compatible in-line barcode. PCR products were cleaned up by NEXTflex cleanup beads, and size selection of the DNAs was performed on a Sage Scientific Blue Pippin. Sequencing was carried out on a 1 × 75 flow cell on the NextSeq 500 platform (Illumina) at the Arizona State University's genomics core and on a 2 × 150 flow cell NovaSeq platform at the genomics core, University of Colorado, Denver.

## Bioinformatics pipelines used for small RNA analysis

Small RNA reads were quality checked using FastQC, and the adapter sequences were cleaved and trimmed using FASTX toolkit. Next, 15–35-nt size reads were mapped to whitefly genome (MEAM1 genome v1.2) using Bowtie with default parameters (Chen et al, 2016). The genome-mapped reads were isolated for the downstream analysis. mirDeep2 was used to annotate the miRNAs (Friedlander et al, 2008). Initial calls by the algorithm were manually inspected for recognized features of miRNAs (Berezikov et al, 2010). Annotations that showed evidence of mature and star strands in the appropriate Dicer cleavage register as well as significant expression were placed in the confident category. Deviation from these characteristics resulted in placement of annotation in the candidate category.

For non-miRNA annotations, small RNA reads, either taking all reads, 19–23-nt sized reads, and 25–30 nt reads were aligned using Bowtie multi-mapping (-a -m 100) options. Bowtie was also used to identify the targets by allowing three mismatches. Size distributions were calculated with basic unix commands: awk, sort, uniq, etc. Using Bowtie alignments ping-pong overlap, piRNA phasing, and Dicer siRNA overhangs signatures were calculated as previously reported (Antoniewski, 2014; Han et al, 2015). SAMtools and bedtools were used to count read alignments and identify high-expressing regions and bias toward short and long read loci, as well as determine potential targets (Quinlan & Hall, 2010). The R packages Scatterplot3d, sushi, heatmap2, pheatmap, and ggplot2 were used to draw the read density graphs (Kolde, 2012; Phanstiel et al, 2014; Warnes et al, 2016; Wickham, 2016; Ligges et al, 2018). The seqlogo program was used to visualize nucleotide biases (Crooks et al, 2004). Read subsetting based on sequence content was carried out using standard Linux tools (grep, awk, etc.).

# Supplementary Information

# Acknowledgements

JK Brown and M Mondal are supported by the Cotton Incorporated Project # 06-829 and USDA-NIFA Project ARZT-3026620-G25-574, GRANT#12469271. A Flynt is supported by NSF MCB 1845978. Computational resources were provided by NSF: ACI 1626217.

## Author Contributions

M Mondal: conceptualization, data curation, formal analysis, validation, investigation, methodology, and writing—original draft.
JK Brown: conceptualization, resources, supervision, funding acquisition, project administration, and writing—review and editing.
A Flynt: conceptualization, resources, data curation, software, formal analysis, supervision, investigation, visualization, methodology, and writing—original draft, review, and editing.

## Conflict of Interest Statement

The authors declare that they have no conflict of interest.

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
