## [Reviewer comments · Life Science Alliance]

Life Science Alliance

Exploiting somatic piRNAs in *Bemisia tabaci* enables a novel gene silencing through RNA feeding

Mosharrof Mondal, Judith Brown, and Alex Flynt
DOI: <https://doi.org/10.26508/lsa.202000731>

Corresponding author(s): Alex Flynt, USM

Review Timeline:

Submission Date:	2020-04-06
Editorial Decision:	2020-04-29
Revision Received:	2020-07-08
Editorial Decision:	2020-07-22
Revision Received:	2020-07-27
Accepted:	2020-07-29

Transaction Report:

April 29, 2020

Re: Life Science Alliance manuscript #LSA-2020-00731-T

Alex Flynt
USM
514 JST
100 Charles Ln dr
Hattiesburg 39406

Dear Dr. Flynt,

Thank you for submitting your manuscript entitled "Exploiting somatic piRNAs in the whitefly enables a new mode of gene silencing through RNA feeding" to Life Science Alliance. The manuscript was assessed by expert reviewers, whose comments are appended to this letter.

As you will see, all reviewers appreciate your work. However, they expect some additional support for the conclusions drawn and provide constructive input on how to provide such support. We think the requests are addressable, especially given the guidance the reviewers provide, and we would thus like to invite you to submit a revised version to us. In our view, the revisions should typically be achievable in around 3 months. However, we are aware that many laboratories cannot function fully during the current COVID-19/SARS-CoV-2 pandemic and therefore encourage you to take the time necessary to revise the manuscript to the extent requested above. We will extend our 'scooping protection policy' to the full revision period required. If you do see another paper with related content published elsewhere, nonetheless contact me immediately so that we can discuss the best way to proceed.

Please note that papers are generally considered through only one revision cycle, so strong support from the referees on the revised version is needed for acceptance.

Thank you for this interesting contribution to Life Science Alliance. We are looking forward to

receiving your revised manuscript.

Sincerely,

B. MANUSCRIPT ORGANIZATION AND FORMATTING:

Reviewer #1 (Comments to the Authors (Required)):

The manuscript by Mondal et al describes a very interesting set of experiments that aims to utilize the piRNA pathway of the whitefly to induce gene silencing by RNA feeding. As a start, the authors present data that broadly describe small RNA and argonaute gene expression in different tissues. Indeed, the piRNA pathway appears to be active also somatically in this insect. Then the authors delve deeper into the si and piRNA pathways and present convincing data that clearly describe how si- and piRNA pathways are intertwined in this species. They then move to feeding flies ssRNA or dsRNA and check if this results in general, non-specific effects. This is not the case. dsRNA feeding, as expected, triggered siRNA production, and excitingly ssRNA could induce silencing when fused to sequences that will be recognized by endogenous piRNAs.

Specific issues:

- Most important point to address is whether the piRNA arms are required on both sides of the target sequence? From a mechanistic point of view, at least based on current knowledge, an arm only on the 3' side should suffice, and an arm on the 5' side should be not functional (due to Zucchini directionality). Ideally this should be addressed by experiment. But maybe there is evidence from the small RNA seq following the feeding that the piRNA signal moves into the target region specifically from the 5' side?
- The figure legends are very brief and hard to understand without also consulting the main text. Please make self-explanatory. What sort of data is displayed? What do the displayed numbers mean? Etc.
- Overall, the text is rather long, especially in introduction and discussion. Some passages are almost review-like, I believe these parts can be written more to-the-point.
- how clean are the dissected tissues? Is there an objective way to judge/measure this? For instance using known miRNA expression data from other insects to identify homologs of highly tissue specific miRNAs?
- Figure 3B is quite small. Is there also a dicer signature right and left of the convergent region (with elevated small RNA counts)?
- How exactly were Dicer signatures calculated? Does this result in a value that represents specificity and/or strength? Also relevant for figure 5 E, F: what is on the (lacking) y-axis?
- Typo in Figure 2 legend (duplicated 'in').

Reviewer #2 (Comments to the Authors (Required)):

In this manuscript by Mondal et al, the authors showed that feeding of exogenous piRNA triggers could potentially function as a new strategy for RNAi technology for pest control in whitefly. First, the authors identified that PIWI family members, as well as piRNAs, are expressed in somatic cell by in-depth analysis of the RNAi pathways in whitefly. Next, the authors found endo-piRNAs in whitefly generated from known siRNA locus based on the small RNA sequencing datasets and computational analysis. They further revealed that both piRNAs and siRNAs are involved in TE silencing and mRNA regulation. Finally, the authors investigated the processing of ingested dsRNA and ssRNA molecules and showed exogenous designed piRNAs could also trigger gene silencing in whitefly as conventional siRNA tools. The application of piRNAs for pest control is interesting and attractive. Following comments may help in further improvement.

Major points:

1. My major concern is that most conclusions excessively relied on bioinformatics analysis and lacked biochemical assays for confirmation. Here, I suggest the authors to furtherly character the features of piRNAs (1U bias in 3' terminal and 2'-O-methylation in 5' terminal) and address the knockdown efficiency of piRNA for target genes (AQP1 and AGLU1) in protein levels.

2. In Figure 1B, x-axis should be expanded to make the data more clearly and y-axis should be segmented into two parts to better show the difference.

3. In Figure 5, the authors observed the signature of ping-pong piRNAs (10 nt overlap) and siRNAs (2 nt overhang) regardless of whether the ssRNA or dsRNA configuration was used. I'm wondering which is mainly functional in down-regulating targets? I suggest the authors compare the ratio of ping-pong piRNA to siRNA.

Minor points:

Page 6, 1st paragraph, "The other mechanism relies on a Piwi protein trafficking to the nucleus where it cleaves designated transcripts that then become substrates for the RNase Zucchini (Zuc)". Based on my knowledge, Piwi in nucleus functions to silence TE at the epigenetic level but not to cleave target transcripts. Piwi-mediated target cleavage should occur in cytoplasm. Moreover, Zucchini is an outer mitochondrial membrane protein and Zucchini-dependent piRNA processing is triggered by the cytoplasmic processing machinery.

Reviewer #3 (Comments to the Authors (Required)):

In this study, the authors set out to characterize whitefly small RNA pathways to use these for targeted pest-control actions using fed small RNA precursors.

The authors computationally identify multiple Argonaute proteins and try to classify them into functional groups. Furthermore, they set out to identify miRNA, siRNA and piRNA loci genome-wide. The focused analysis of si and piRNA loci reveals that a large fraction overlaps and that probably the piRNA pathway is also active widespread in somatic cells. In the end, the authors set out to explore the potential use of the somatic piRNA pathway for gene knock-downs as an alternative/addition to the siRNA pathway by feeding RNA precursors. This last part is the most exciting part of the manuscript but unfortunately also the least convincing part of the manuscript. In the study, the authors try to test the effect of feeding either precursors for the siRNA or piRNA pathway and measure the effects on the target mRNAs. Whereas both precursors show the expected effect to varying extent the data presented falls short in providing evidence that the effect when feeding piRNA precursors is indeed mediated via this pathway. The presented experiments cannot rule out that the observed effect is solely triggered by the siRNA pathways. Additional control-experiments would be required to support the claim of the authors that indeed the piRNA pathway contributes to the gene silencing effect when feeding exogenous precursor RNAs.

Comments:

- The explanation of piRNA biogenesis modes in *Drosophila melanogaster* in the introduction is not correct. While it is true that ping-pong is a system of piRNA amplification induced by slicing, there is no evidence that the slicing activity of the protein Piwi is required. It is merely the proteins Aub and AGO3 that are forming the ping-pong pair. These two have the ability of slicing target RNAs depending on complementarity with the loaded piRNA. This generates the 5' end of a new piRNA that will be loaded into one of the PIWI proteins. The 3' end of the new piRNA can be formed by multiple pathways. If the 3' end is generated by the endonuclease Zucchini, this generates a new 5' end in addition that gets predominantly loaded into the protein Piwi with another Zucchini cleavage event generating the 3' of this piRNA and yet again a 5' end. This process can continue for a certain stretch of sequence generating the so called phased piRNAs. This is the pathway in the *Drosophila* germline. In the somatic follicle cells only phased piRNAs are observable that are generated by a similar phasing process. In this case the initiating event is unknown due to the absence of slicing Argonaute proteins. There is no evidence for target cleavage and therefore

precursor definition by the protein Piwi in the cytoplasm or the nucleus.

- Along similar lines, the presence of Zuc and Armitage alone cannot predict the presence of a phasing piRNA pathway. For example, the silkworm *Bombyx mori* only shows ping-pong activity and no phased piRNA production, yet Zucchini and Armitage are not only present but also required for faithful piRNA production. To solidify the results on ping-pong and phased piRNA production pathways additional analyses would be ideal. For example:

- o Do piRNAs (or all small RNAs) in whitefly show a 1 U pattern?

- o If yes, the pattern of 1U to 10A could serve as a refinement of the ping-pong analysis. In the literature it is typically required that ping-pong piRNAs not only overlap by 10 nt but also that they show the expected nucleotide pattern.

- o For the identification of phased piRNAs an analysis studying the coupling of piRNA 3' and 5' ends can be performed.

- The classification of Argonaute proteins and initial sRNA characterization seems solid and provides a good first insight into the small RNA biology to be expected in this organism.

- The identification and classification of si and piRNA loci is interesting. One point that is lacking in my opinion is the incorporation of ping-pong sources. At this stage of the manuscript only single-stranded phased piRNA producing source loci are reported. In contrast later on when endogenous fed precursor generated small RNAs are analyzed only ping-pong piRNAs are considered. It would be good to consider both classes of piRNAs in both stages of the manuscript.

- The analysis of fed dsRNAs from a different species does support all the claims the authors raise and serve as a good and controlled entry into the last section of the manuscript. It would be interesting to provide some analyses from this section also for the final experiments. In this section partly degradation of the fed dsRNA precursor was observed. Was such a degradation also observed in the later experiments using ss or dsRNA containing the piRNA inducing sequences? Typically, ssRNA is considered to be more unstable than dsRNA, was something like this observable in the data?

- While a very interesting approach that could extend the use of small RNA pathways for pest-control, the analysis of the experimental gene knockdown with fed RNAs has some shortcomings (see below). At the moment it is very difficult to support all of the claims the author's raised with the presented data. The QPCR data suggests that ssRNAs as precursors for piRNA biogenesis are nearly as potent as fed dsRNA and the authors attribute this effect to the piRNA pathway. But when the authors analyze the produced small RNAs in detail it gets very unclear which class of small RNA elicits these effects.

- o From the methods it is unclear which strand was generated for the ssRNA feeding. As a gene-knockdown response was mounted it must have been the sequence complementary to the mRNA

- o This could also explain the generation of siRNAs from the single stranded precursor (as also briefly mentioned in the discussion). The precursor could hybridize with the mRNA present in the cells generating a dsRNA that can be used by siRNA machinery. This presence of siRNAs in all experiments makes it quite difficult to evaluate the role of the piRNA pathway in the silencing response.

- o In Fig. 5 E/F/H/I it seems that comparable levels of siRNAs and piRNAs have been observed in both the ssRNA and dsRNA experiments. To evaluate produced small RNAs better it would be helpful to show

- Size-profiles would be very helpful in the evaluation of the small RNA results.

- All small RNAs and not only small RNAs with dicer or ping-pong signature

- Are Y-axes in these graphs the same? From the plots in G and J it seems that the dual-stranded precursor generates more small RNAs, yet the graphs in E/F/H/I do not show this difference

- o To fully understand the small RNAs responsible for the observed gene knock-down effect additional controls would be required:

- Clean dsRNA containing the piRNA trigger sequences (the T7 dual-sided transcription probably

does not produce clean dsRNA. Is it possible that this is the reason why you observe piRNA production from these dsRNA precursors?)

- ssRNA with mRNA target sequence in sense orientation towards the mRNA (no siRNA production should occur by simple hybridization)
- ssRNA without any piRNA arms. In this case no piRNAs should be produced
 - do you still get siRNAs?
 - Are you still able to observe a gene knock-down effect?
- Did you consider using a locus producing phased piRNAs for your piRNA constructs?

Overall, I find the study and the practical use of piRNAs for pest controls very interesting, but the manuscript would require additional analyses and controls to support the presented claims.

Reviewer #1 (Comments to the Authors (Required)):

The manuscript by Mondal et al describes a very interesting set of experiments that aims to utilize the piRNA pathway of the whitefly to induce gene silencing by RNA feeding. As a start, the authors present data that broadly describe small RNA and argonaute gene expression in different tissues. Indeed, the piRNA pathway appears to be active also somatically in this insect. Then the authors delve deeper into the si and piRNA pathways and present convincing data that clearly describe how si- and piRNA pathways are intertwined in this species. They then move to feeding flies ssRNA or dsRNA and check if this results in general, non-specific effects. This is not the case. dsRNA feeding, as expected, triggered siRNA production, and excitingly ssRNA could induce silencing when fused to sequences that will be recognized by endogenous piRNAs.

Specific issues:

-Most important point to address is whether the piRNA arms are required on both sides of the target sequence? From a mechanistic point of view, at least based on current knowledge, an arm only on the 3' side should suffice, and an arm on the 5' side should be not functional (due to Zucchini directionality). Ideally this should be addressed by experiment. But maybe there is evidence from the small RNA seq following the feeding that the piRNA signal moves into the target region specifically from the 5' side?

Thank you for the comment. We have extensively reanalyzed our data to provide greater insight into the patterns of piRNA biogenesis associated with the fed RNAs. In particular by contrasting the differences between the manner in which the two triggers were cloned we were able to produce salient insights that address the concern raised by this comment. The two triggers we fed animals were clone either with the sense strand of the piRNA locus (AQP1) or the antisense-strand (AGLU1). The major difference we find is that the sense version is recognized by the endogenous pathways as a designated phasing strand. Simultaneously, when only a single strand is provided this can become phased as well. Importantly, we see that phased piRNA production can be produced from internal parts of endogenous locus, suggesting that activation of phasing biogenesis is more permissive in whitefly. This finding is a major highlight of the study, and will be key to the successful design of the piRNA cassette.

-The figure legends are very brief and hard to understand without also consulting the main text. Please make self-explanatory. What sort of data is displayed? What do the displayed numbers mean? Etc.

We have revised the figure legends.

-Overall, the text is rather long, especially in introduction and discussion. Some passages are almost review-like, I believe these parts can be written more to-the-point.

We have revised the main text of the manuscript and made the suggested changes.

-how clean are the dissected tissues? Is there an objective way to judge/measure this? For

instance, using known miRNA expression data from other insects to identify homologs of highly tissue specific miRNAs?

Dissected whitefly tissue/organs were extirpated using sterile technique and nuclease-free PURE water or autoclaved buffer. The insects were dissected following a published protocol from Brown lab. We have added the reference in the main text, that provides the verification of library *Joseph M. Cicero, Judith K. Brown, Anatomy of Accessory Salivary Glands of the Whitefly Bemisia tabaci (Hemiptera: Aleyrodidae) and Correlations to Begomovirus Transmission, Annals of the Entomological Society of America, Volume 104, Issue 2, 1 March 2011, Pages 280–286, <https://doi.org/10.1603/ANI10171>*

-Figure 3B is quite small. Is there also a Dicer signature right and left of the convergent region (with elevated small RNA counts)?

We have combined figure 3B and 3C in this new version of the manuscript. The new figure 3B top panel: Dicer signature containing reads can be seen on both sides of the convergent region. However, the density of the reads is low compared to the convergent region. However, it is not unexpected to find putative siRNAs as we show co-occurrence of siRNAs and piRNAs is widespread in whiteflies

-How exactly were Dicer signatures calculated? Does this result in a value that represents specificity and/or strength? Also relevant for figure 5 E, F: what is on the (lacking) y-axis?

The details of the Dicer and ping-pong signature identification/calculation can be found in this paper:

Antoniewski C. Computing siRNA and piRNA overlap signatures. Methods Mol Biol. 2014;1173:135-146. doi:10.1007/978-1-4939-0931-5_12

Using these tools reads with designated overlaps can be isolated from alignments. For siRNAs we determine reads that overlap by 2 less than their length. This simulates Dicer cleavage patterns. In the revision we provide a new type of analysis in Fig 2C that shows in greater detail how small RNA pairs can be matched to reveal duplexes produced by Dicer.

Figure 5 in the initial submission is no longer part of the manuscript. There are now two figures (5&6) that a more rational approach to understanding the small RNA varieties produced by the piRNA-based RNAi triggers.

-Typo in Figure 2 legend (duplicated 'in').

Fixed

Reviewer #2 (Comments to the Authors (Required)):

In this manuscript by Mondal et al, the authors showed that feeding of exogenous piRNA triggers could potentially function as a new strategy for RNAi technology for pest control in

whitefly. First, the authors identified that PIWI family members, as well as piRNAs, are expressed in somatic cell by in-depth analysis of the RNAi pathways in whitefly. Next, the authors found endo-piRNAs in whitefly generated from known siRNA locus based on the small RNA sequencing datasets and computational analysis. They further revealed that both piRNAs and siRNAs are involved in TE silencing and mRNA regulation. Finally, the authors investigated the processing of ingested dsRNA and ssRNA molecules and showed exogenous designed piRNAs could also trigger gene silencing in whitefly as conventional siRNA tools. The application of piRNAs for pest control is interesting and attractive. Following comments may help in further improvement.

Major points:

1. My major concern is that most conclusions excessively relied on bioinformatics analysis and lacked biochemical assays for confirmation. Here, I suggest the authors to furtherly character the features of piRNAs (1U bias in 3' terminal and 2'-O-methylation in 5' terminal) and address the knockdown efficiency of piRNA for target genes (AQP1 and AGLU1) in protein levels.

We appreciate the reviewer's concern and have made extensive use of 1U read identities for characterizing small RNAs. 2'-O-methyl analysis would not allow us to distinguish siRNAs from piRNAs, while comparing read length does, thus 2'-O-methyl analysis was not included. To address this concern as part of the revision we have revised the entire bioinformatic analysis and added deep characterization of small RNA sequencing. All the main figures have been improved with results from the new analysis. We hope this sufficiently addresses reviewer's point, that further characterization of piRNAs is needed. Moreover, we hope that the reviewer will also see the value in extensive analysis of standard small RNA seq datasets to characterize RNAi pathways in a non-model organism. When investigating biology in such a setting tools and protocols are absent. The approaches we used would apply to any species, suggesting this article could serve as a rubric for such a study. Along these lines, reagents like antibodies are unavailable for the whitefly, making protein quantification challenging. RNA quantification to assess gene silencing is a standard approach for agricultural pest research.

2. In Figure 1B, x-axis should be expanded to make the data more clearly and y-axis should be segmented into two parts to better show the difference.

The figure is revised, and necessary changes have been made

3. In Figure 5, the authors observed the signature of ping-pong piRNAs (10 nt overlap) and siRNAs (2 nt overhang) regardless of whether the ssRNA or dsRNA configuration was used. I'm wondering which is mainly functional in down-regulating targets? I suggest the authors compare the ratio of ping-pong piRNA to siRNA.

As part of the *in toto* revision of computational approaches we have identified piRNA and siRNA characteristics which were used to identify differences in ssRNA and dsRNA triggers. We hope the reviewer will be satisfied by our finding that ssRNA triggers produce mainly piRNAs and while dsRNA version produce a mix of siRNAs and piRNAs. From this we were able to conclude that gene silencing occurred through piRNAs. We took this one step further by characterizing the contribution of ping-pong and phasing piRNA sub-types.

Minor points:

Page 6, 1st paragraph, "The other mechanism relies on a Piwi protein trafficking to the nucleus where it cleaves designated transcripts that then become substrates for the RNase Zucchini (Zuc)". Based on my knowledge, Piwi in nucleus functions to silence TE at the epigenetic level but not to cleave target transcripts. Piwi-mediated target cleavage should occur in cytoplasm. Moreover, Zucchini is an outer mitochondrial membrane protein and Zucchini-dependent piRNA processing is triggered by the cytoplasmic processing machinery.

We thank the reviewer for noticing this inaccuracy. The introduction has been revised.

Reviewer #3 (Comments to the Authors (Required)):

In this study, the authors set out to characterize whitefly small RNA pathways to use these for targeted pest-control actions using fed small RNA precursors.

The authors computationally identify multiple Argonaute proteins and try to classify them into functional groups. Furthermore, they set out to identify miRNA, siRNA and piRNA loci genome-wide. The focused analysis of si and piRNA loci reveals that a large fraction overlaps and that probably the piRNA pathway is also active widespread in somatic cells. In the end, the authors set out to explore the potential use of the somatic piRNA pathway for gene knock-downs as an alternative/addition to the siRNA pathway by feeding RNA precursors. This last part is the most exciting part of the manuscript but unfortunately also the least convincing part of the manuscript. In the study, the authors try to test the effect of feeding either precursors for the siRNA or piRNA pathway and measure the effects on the target mRNAs. Whereas both precursors show the expected effect to varying extent the data presented falls short in providing evidence that the effect when feeding piRNA precursors is indeed mediated via this pathway. The presented experiments cannot rule out that the observed effect is solely triggered by the siRNA pathways. Additional control-experiments would be required to support the claim of the authors that indeed the piRNA pathway contributes to the gene silencing effect when feeding exogenous precursor RNAs.

Comments:

- The explanation of piRNA biogenesis modes in *Drosophila melanogaster* in the introduction is not correct. While it is true that ping-pong is a system of piRNA amplification induced by slicing, there is no evidence that the slicing activity of the protein Piwi is required. It is merely the proteins Aub and AGO3 that are forming the ping-pong pair. These two have the ability of slicing target RNAs depending on complementarity with the loaded piRNA. This generates the 5' end of a new piRNA that will be loaded into one of the PIWI proteins. The 3' end of the new piRNA can be formed by multiple pathways. If the 3' end is generated by the endonuclease Zucchini, this generates a new 5' end in addition that gets predominantly loaded into the protein Piwi with another Zucchini cleavage event generating the 3' of this piRNA and yet again a 5' end. This process can continue for a certain stretch of sequence generating the so called phased piRNAs. This is the pathway in the *Drosophila* germline. In the somatic follicle cells only phased piRNAs are observable that are generated by a similar phasing process. In this case the initiating event is unknown due to the absence of slicing Argonaute proteins. There is no evidence for

target cleavage and therefore precursor definition by the protein Piwi in the cytoplasm or the nucleus.

We thank the reviewer for noticing this inaccuracy. The introduction has been revised.

- Along similar lines, the presence of Zuc and Armitage alone cannot predict the presence of a phasing piRNA pathway. For example, the silkworm *Bombyx mori* only shows ping-pong activity and no phased piRNA production, yet Zucchini and Armitage are not only present but also required for faithful piRNA production. To solidify the results on ping-pong and phased piRNA production pathways additional analyses would be ideal. For example:

- o Do piRNAs (or all small RNAs) in whitefly show a 1 U pattern?

In the revision we have revised the computational approaches. A major difference is the inclusion of assessing phasing through calculating proximity of trailing 1U reads. Using this method, we see phasing piRNAs are widespread in whitefly. This is seen in other Hemipterans as phasing piRNAs are also present in Aphids (ref: Gainetdinov, et al., (2018))

- o If yes, the pattern of 1U to 10A could serve as a refinement of the ping-pong analysis. In the literature it is typically required that ping-pong piRNAs not only overlap by 10 nt but also that they show the expected nucleotide pattern.

We indeed see this pattern, and have applied it in several places in the analysis (Figure 1, Figure 5, Figure 6). However, bulk piRNAs in whitefly seem to be produced by phasing, thus the 10A signature is minor relative to 1U. Please see inline diagram.

- o For the identification of phased piRNAs an analysis studying the coupling of piRNA 3' and 5' ends can be performed.

This has been included throughout the manuscript.

- The classification of Argonaute proteins and initial sRNA characterization seems solid and provides a good first insight into the small RNA biology to be expected in this organism.
- The identification and classification of si and piRNA loci is interesting. One point that is lacking in my opinion is the incorporation of ping-pong sources. At this stage of the manuscript only single-stranded phased piRNA producing source loci are reported. In contrast later on when endogenous fed precursor generated small RNAs are analyzed only ping-pong piRNAs are considered. It would be good to consider both classes of piRNAs in both stages of the manuscript.

In Figure 2, we analyze all loci rather than a subset. Phasing analysis is a major feature of this. We hope the reviewer will particularly be satisfied by how we considered both modes of piRNA biogenesis when evaluating small RNA production from the piRNA triggers.

- The analysis of fed dsRNAs from a different species does support all the claims the authors raise and serve as a good and controlled entry into the last section of the manuscript. It would be interesting to provide some analyses from this section also for the final experiments. In this section partly degradation of the fed dsRNA precursor was observed. Was such a degradation also observed in the later experiments using ss or dsRNA containing the piRNA inducing sequences? Typically, ssRNA is considered to be more unstable than dsRNA, was something like this observable in the data?

As proposed by the reviewer, degradation is the fate of many fed RNAs. The suggestion was so helpful that it led us to develop computational approaches to isolate functional small RNAs from within a collection of degradation products. Please see revisions to Figure 4, 5 and the new Figure 6.

- While a very interesting approach that could extend the use of small RNA pathways for pest-control, the analysis of the experimental gene knockdown with fed RNAs has some shortcomings (see below). At the moment it is very difficult to support all of the claims the author's raised with the presented data. The QPCR data suggests that ssRNAs as precursors for piRNA biogenesis are nearly as potent as fed dsRNA and the authors attribute this effect to the piRNA pathway. But when the authors analyze the produced small RNAs in detail it gets very unclear which class of small RNA elicits these effects.

As part of addressing concerns in the previous comment we developed methods to identify bona fide small RNAs among degradation products. This allowed us to observe that siRNAs are not a major product from ssRNAs, while they are much more plentiful with dsRNA triggers. Now we can positively identify piRNAs as effectors of gene silencing.

- o From the methods it is unclear which strand was generated for the ssRNA feeding. As a gene-knockdown response was mounted it must have been the sequence complementary to the mRNA

The ssRNA triggers were transcribed to generate RNAs complementary to the targets AQP1 and AGLU1. We did notice that the orientation of the piRNA trigger had been flipped to a quirk of the cloning procedure. However, this allowed us to gain even greater insight into the biogenesis of piRNAs from the triggers and provide rules for effective gene silencing through this approach.

- o This could also explain the generation of siRNAs from the single stranded precursor (as also briefly mentioned in the discussion). The precursor could hybridize with the mRNA present in the cells generating a dsRNA that can be used by siRNA machinery. This presence of siRNAs in all experiments makes it quite difficult to evaluate the role of the piRNA pathway in the silencing response.

The revised analysis finds fewer siRNAs and more ping-pong piRNAs, which are likely derived from target mRNAs. This is discussed in the results and discussion sections.

o In Fig. 5 E/F/H/I it seems that comparable levels of siRNAs and piRNAs have been observed in both the ssRNA and dsRNA experiments. To evaluate produced small RNAs better it would be helpful to show

- ♣ Size-profiles would be very helpful in the evaluation of the small RNA results.
- ♣ All small RNAs and not only small RNAs with dicer or ping-pong signature
- ♣ Are Y-axes in these graphs the same? From the plots in G and J it seems that the dual-stranded precursor generates more small RNAs, yet the graphs in E/F/H/I do not show this difference

Figure 5 is completely revised, and now there is a Figure 6. Size profiles are included. Read quantifications are included in all plots of read density.

o To fully understand the small RNAs responsible for the observed gene knock-down effect additional controls would be required:

- ♣ Clean dsRNA containing the piRNA trigger sequences (the T7 dual-sided transcription probably does not produce clean dsRNA. Is it possible that this is the reason why you observe piRNA production from these dsRNA precursors?)

The whitefly gut is inhospitable to RNA, as it evident from the degradation fragments we observe. It is likely that even a clean dsRNA would be unwound in this environment

- ♣ ssRNA with mRNA target sequence in sense orientation towards the mRNA (no siRNA production should occur by simple hybridization)

We address this concern regarding the recovery of apparent siRNAs from the single stranded precursor. By using a refined analysis, we have found siRNAs to be a minor population in single stranded triggers. Our prior analysis incorrectly included degradation fragments in estimates of siRNA abundance. We hope the reviewer will find the dissection of siRNA, and piRNA types in Figure 6 satisfying

- ♣ ssRNA without any piRNA arms. In this case no piRNAs should be produced
- do you still get siRNAs?
- Are you still able to observe a gene knock-down effect?

Here again, the reviewer is asking us to explore the apparent siRNAs that we reported from single-stranded triggers. The new analysis resolves this by demonstrating siRNAs are not a major product, as was initially presented, for the single stranded trigger

- ♣ Did you consider using a locus producing phased piRNAs for your piRNA constructs?

The more detailed analysis we provide indicate that the loci used to trigger piRNAs have a phasing signature. We hope the reviewer will find the behaviors we identify for triggers in different orientations represent a significant advance.

July 22, 2020

RE: Life Science Alliance Manuscript #LSA-2020-00731-TR

Dr. Alex Flynt
USM
514 JST
100 Charles Ln dr
Hattiesburg 39406

Dear Dr. Flynt,

Thank you for submitting your revised manuscript entitled "Exploiting somatic piRNAs in *Bemisia tabaci* enables a novel gene silencing through RNA feeding". Your manuscript was re-reviewed by the original referees, and their reports are attached below. We would be happy to publish your paper in Life Science Alliance pending final revisions necessary to meet our formatting guidelines.

- Please consider the remaining point of referee #3 regarding Figure 6 and adjust the representation related discussion accordingly
- please add the author contributions and conflict of interest statement to the main manuscript text
- please upload the main and supplementary figures as single files
- please add your figure legends as a separate section in the main manuscript text
- please add a figure legend for Fig. S3
- please upload tables as editable doc or excel files
- please upload your manuscript as an editable doc file
- please list 10 authors et al. in your references

A. FINAL FILES:

B. MANUSCRIPT ORGANIZATION AND FORMATTING:

Sincerely,

Reilly Lorenz
Editorial Office Life Science Alliance
Meyerhofstr. 1
69117 Heidelberg, Germany
t +49 6221 8891 414
e contact@life-science-alliance.org
www.life-science-alliance.org

Reviewer #2 (Comments to the Authors (Required)):

In the revised version, my major concerns about the features of piRNAs have been fully addressed. I have no further request for modifications.

Reviewer #3 (Comments to the Authors (Required)):

In this study, the authors set out to characterize whitefly small RNA pathways to use these for targeted pest-control actions using fed small RNA precursors. The authors computationally identify multiple Argonaute proteins and try to classify them into functional groups. Furthermore, they set out to identify miRNA, siRNA and piRNA loci genome-wide. The focused analysis of si and piRNA loci reveals that a large fraction overlaps and that probably the piRNA pathway is also active widespread in somatic cells. In the end, the authors set out to explore the potential use of the somatic piRNA pathway for gene knock-downs as an alternative/addition to the siRNA pathway by feeding RNA precursors.

Overall the manuscript improved to quite some extent. The text is still very long and written not to the point but this can be solved by careful rephrasing.

It seems that the reviewer suggestions have been incorporated carefully into the manuscript which makes the analysis more robust. There are quite some new and improved computational analyses that support the claims. Especially adding analyses for 1U and more details ping-pong and siRNA identifications make the data clearer. The weak point is still the ending of the manuscript.

Unfortunately the new figure 6 is over-complicated and the discussion lacks alternative explanations of the data in addition to the very strong claim the authors make.

Additional Comments:

Figure #6 is very complicated and difficult to understand. Also, the discussion in the results section is still quite complicated and very strong for the data underlying these claims. What do you mean with internal initiation? In Fig6 you can observe ping-pong piRNAs right from the beginning of piRB6 arm. Also, you do not take into account slicing events that do not form a pair. Potentially you have abundant slicing of piRB6 by an argonaute protein (or cleavage by other means) that induce the phased piRNA production. This could be similar to the Drosophila somatic follicle cells where only phased piRNAs exist and they are initiated without any slicing event of an Argonaute protein.

July 29, 2020

RE: Life Science Alliance Manuscript #LSA-2020-00731-TRR

Dr. Alex Flynt
USM
514 JST
100 Charles Ln dr
Hattiesburg 39406

Dear Dr. Flynt,

Thank you for submitting your Research Article entitled "Exploiting somatic piRNAs in *Bemisia tabaci* enables a novel gene silencing through RNA feeding". It is a pleasure to let you know that your manuscript is now accepted for publication in Life Science Alliance. Congratulations on this interesting work.

DISTRIBUTION OF MATERIALS:

Again, congratulations on a very nice paper. I hope you found the review process to be constructive and are pleased with how the manuscript was handled editorially. We look forward to future exciting submissions from your lab.

Sincerely,

Reilly Lorenz
Editorial Office Life Science Alliance
Meyerhofstr. 1
69117 Heidelberg, Germany
t +49 6221 8891 414
e contact@life-science-alliance.org
www.life-science-alliance.org